# Gender Expression, Weight Status, and Risk of Experiencing Eating Disorders among Gender-Diverse Adults Assigned Male at Birth in Bangkok, Thailand

**DOI:** 10.3390/nu15173700

**Published:** 2023-08-24

**Authors:** Thanit Vinitchagoon, Waris Wongpipit, Phenphop Phansuea

**Affiliations:** 1Food and Nutrition Academic and Research Cluster, Institute of Nutrition, Mahidol University, Nakhon Pathom 73170, Thailand; thanit.vin@mahidol.edu; 2Department of Curriculum and Instruction, Faculty of Education, Chulalongkorn University, Bangkok 10330, Thailand; waris.w@chula.ac.th; 3Department of Sports Science and Physical Education, Faculty of Education, The Chinese University of Hong Kong, Hong Kong SAR, China

**Keywords:** gender, LGBTQ+, eating disorders, weight status

## Abstract

This study examines the association between gender expression, weight status, and the risk of experiencing eating disorders among gender-diverse adults assigned male at birth living in Bangkok, Thailand. Participants completed self-administered questionnaires to provide demographic data and anthropometric measures, and an Eating Attitude Test-26 (EAT-26) to determine the risk of experiencing eating disorders. The associations between gender expression, weight status, and the risk of experiencing eating disorders were analyzed using multivariable logistic regression models. No significant differences were observed in weight-related variables based on gender expression. Participants self-described as feminine/androgynous had lower odds of experiencing a high risk of eating disorders compared to those self-described as masculine (odds ratio (OR) = 0.49; 95% confidence interval (95% CI) = 0.27, 0.88). A higher body mass index (BMI) (OR = 1.07; 95% CI = 1.01, 1.14) and BMI discrepancy (OR = 1.13; 95% CI = 1.03, 1.24) were associated with higher odds of engaging in extreme weight-control behaviors. The risk of experiencing eating disorders among Thai gender-diverse adults assigned male at birth could differ across gender expression and weight status. Further research is needed to expand the understanding of these relationships and develop tailored intervention programs to mitigate the risk.

## 1. Introduction

The prevalence of eating disorders and disordered eating behaviors is higher among gender-diverse individuals, often described as lesbian, gay, bisexual, transgender, queer, and/or questioning (LGBTQ+), compared to their cisgender heterosexual counterparts [1]. Unique risk factors contributing to the development of eating disorders in this gender-diverse population include stigma, prejudice, and discrimination towards revealing gender identity and sexual orientation [2]. Experiencing these risk factors can lead to increased stress, which negatively affects physical and mental health [3].

Within the LGBTQ+ community, subgroups are categorized based on gender identity, expression, and sex assigned at birth. Previous studies have shown variations in the prevalence of eating disorders and disordered eating behaviors among these subgroups based on gender identity, expression, and sex assigned at birth [1]. For instance, individuals assigned female at birth who identify as gender-nonconforming have a higher risk of experiencing eating disorders compared to those who have transitioned from male to female [4]. On the other hand, individuals assigned male at birth are less likely than females to be diagnosed with anorexia nervosa or bulimia nervosa [5]. Due to this complexity, many studies in the gender-diverse population have specified the study population precisely, such as by sex assigned at birth, gender identity, and sexual orientation [6,7]. Thus, it is important to conduct specific studies that focus on different subgroups within the LGBTQ+ population, rather than treating it as a homogeneous group.

Weight-related concerns, along with the associated stigma and discrimination, represent significant additional risk factors for the development of eating disorders [8]. Gender-diverse individuals are more likely to experience weight stigma and discrimination in comparison to cisgender heterosexual populations [9]. The correlation between weight stigma and eating disorders can be explained by the impact of weight bias internalization and resultant psychological distress [10]. The internalization of weight bias is linked to maladaptive eating behaviors and a diminished mental health-related quality of life, observable in both LGBTQ+ and heterosexual cisgender adults [11]. Notably, gay men have exhibited heightened levels of fatphobia, depression, and eating-related concerns when compared to cisgender heterosexual men, underscoring a susceptibility to cognitive biases in relation to weight status [12].

In Thailand, the gender-diverse population continues to encounter substantial challenges rooted in social structures, cultural norms, values, and healthcare laws, which result in unequal opportunities when compared to their cisgender heterosexual counterparts. As of 2020, Thailand’s ranking of 40th out of 175 countries in terms of LGBTQ+ acceptance underscores the evident need for advancements in this area [13]. Furthermore, a paucity of research is dedicated to health concerns within the gender-diverse population, and even the statistics pertaining to the LGBTQ+ demographic in the country remain approximations (approximately 5 million individuals or 8% of the total population as of 2018) [14]. Nonetheless, investigations conducted among gender-diverse communities in Thailand have unveiled parallel issues akin to those encountered in other nations. These encompass challenges related to the self-disclosure of sexual orientation and gender identity, as well as the prevalence of stigma and discrimination, particularly among students and young adults [15]. Notably, no published study to date has delved into the correlation between weight status and eating disorders within the gender-diverse population in Thailand, irrespective of the specific subgroups encompassed by the LGBTQ+ community. The preexisting body of research concerning eating disorders has primarily centered around children and young adults assigned female at birth.

Therefore, the objective of this study was to examine the association between gender expression, weight status, and the risk of experiencing eating disorders among gender-diverse adults assigned male at birth living in Bangkok, Thailand. This study could be a pioneering investigation in this under-investigated area of literature in Thailand and could provide preliminary findings that could lead to further hypotheses and potential interventions to address eating disorders in this population.

## 2. Materials and Methods

### 2.1. Study Design and Ethics Approval

This study drew upon data derived from the Dieting, Exercise, and Eating-Related Behaviors (DEER) Study, a cross-sectional observational investigation conducted within the demographic of gender-diverse adults assigned male at birth, residing in Bangkok, Thailand. This study’s methodology utilized cross-sectional comparisons between two distinct subgroups based on gender expression (masculine vs. feminine/androgynous) within the delineated sample population of gender-diverse adults assigned male at birth. The process of data collection unfolded between the months of August and October in the year 2021, employing online self-administered questionnaires as the primary instrument. To assemble participants, recruitment materials were strategically disseminated through diverse online social media platforms such as Facebook, Twitter, and Instagram. Interested individuals completed a screening survey, and those determined eligible based on the screening results were contacted by the research team. Participants provided electronic consent to participate and subsequently completed a series of self-administered questionnaires. The research team reviewed the data and contacted participants for clarification if the data was determined as unreliable. The research team also contacted participants in cases of missing data.

This study received approval for exemption from the Institutional Review Board (IRB) of Mahidol University (Protocol Number: MU-CIRB 2022/196.2107) for research involving the use of survey procedures (COE No. MU-CIRB 2022/123.0908; Date of Approval: 9 August 2022). The data supporting the findings of this study are available from the corresponding author upon reasonable request.

### 2.2. Participants and Recruitment Procedures

Participants who met the following criteria were included in the study: assigned male at birth, Thai nationality, at least 18 years old at the time of consent, self-identified as LGBTQ+, and resided in Bangkok or nearby provinces while regularly working in Bangkok for a minimum of six months at the time of consent.

Participants were excluded from the study if they did not sign the consent form, expressed unwillingness to complete the questionnaires, experienced mental health or cognitive issues that hindered their ability to complete the questionnaires effectively and independently, or had physical health issues that prevented them from adequately consuming foods by mouth.

### 2.3. Questionnaire and Data Collection

For data collection, self-administered questionnaires were utilized in this study. These questionnaires were specifically designed and managed using the REDcap^®^ platform (Nashville, TN, USA), a secure and web-based software system designed to facilitate online questionnaire administration for research studies [16].

Regarding demographic characteristics, participants provided self-reported information on various aspects including their age (in years), birthplace (province), monthly income (in Thai Baht per month), living status (alone/with friends or family/with partner or couple), religion (atheist/Buddhist/other specified religion), educational attainment (specified level of education), relationship status (single/partnered/married), smoking status (yes/no), alcohol consumption (yes/no), and history of mental health issues (yes/no).

Anthropometric variables were also collected through self-report. Participants provided their weight and height, which were then used to calculate their body mass index (BMI). The BMI values were further categorized into specific weight categories based on the cutoffs recommended by the World Health Organization [17]. Additionally, participants reported their ideal body weight, which refers to the weight they desired to be. The ideal BMI was then calculated based on this ideal body weight. To measure BMI discrepancy, the current BMI was subtracted from the ideal BMI, which resulted in a numerical value representing the difference between the participants’ current and ideal BMI.

Eating disorder-related variables were assessed using the Eating Attitude Test-26 (EAT-26) [18], a widely used standardized measure to evaluate symptoms and concerns associated with eating disorders [19]. This questionnaire serves as a screening tool to identify individuals at risk of experiencing eating disorders. The total score for the EAT-26 ranges from 0 to 78 points and includes three subscales: dieting (0–39 points), bulimia and food preoccupation (0–18 points), and oral control (0–21 points). A total score of 20 points or higher indicates a potential risk of experiencing eating disorders. Additionally, the questionnaire includes four behavioral questions aimed at assessing the presence and frequency of extreme weight-control behaviors.

The EAT-26 has been translated and validated in several populations including the LGBTQ+ community, specifically among individuals identified as gay males [20,21,22]. The EAT-26 was translated into Thai and validated with a sample of Thai young adults assigned female at birth who were diagnosed with eating disorders, with a cutoff point of 12 points or higher determining the risk of eating disorders (sensitivity = 88.6% and specificity = 88.7%) [23]. This cutoff point aligns with another study suggesting that a cutoff point of 11 points or higher could enhance diagnostic accuracy and reduce the likelihood of false negative results [24]. Thus, for this study, a cutoff point of 12 points or higher was used to identify individuals at risk of experiencing eating disorders.

To assess gender expression, we employed the Sexual Orientation and Gender Identity (SOGI) questionnaire [25], which was translated into Thai and validated among Thai adults, including the LGBTQ+ community. The questionnaire demonstrated excellent content validity with a content validity index of 1.00. Participants self-reported their gender expression as masculine, feminine, or androgynous, representing an equal combination of both masculine and feminine traits. We combined the two groups “feminine” and “androgynous” to a single group vs. masculine due to inadequate statistical power to analyze androgynous as a separate group. Also, in this study, participants self-described as “androgynous” reported more traits and characteristics of femininity compared to masculinity.

### 2.4. Statistical Analysis

Several demographic variables were transformed to facilitate regression analyses and enhance the interpretation of the results. Income was transformed into an ordinal variable by dividing it into quartiles due to the non-normal distribution of the data. Additionally, the following variables were transformed into binary variables: birthplace (Bangkok/other provinces), educational attainment (lower than bachelor’s degree/bachelor’s degree or higher), relationship status (single/partnered or married), and gender expression (masculine/feminine or androgynous).

Descriptive statistics were utilized to present the demographic characteristics and weight-related variables of the participants. Continuous variables were reported as means (standard deviation), while categorical variables were reported as frequency (percentage). Independent t-tests were employed to compare mean differences in continuous variables including age, current BMI, ideal BMI, and BMI discrepancy based on gender expression. Chi-square tests were utilized to compare proportional differences in the categorical variables including birthplace, income per month, current living status, religion, educational attainment, relationship status, current smoking status, current alcohol consumption, history of mental health issues, and BMI category based on gender expression.

To investigate the associations between gender expression, BMI, and BMI discrepancy as predictor variables, and the total and subscale scores of EAT-26 as outcome variables, multivariable linear regression was employed. Similarly, multivariable logistic regression was employed to examine the associations between gender expression, BMI, and BMI discrepancy as predictor variables, and the odds of experiencing a high risk for eating disorders (defined as a total EAT-26 score of 12 points or higher) and the odds of engaging in extreme weight-control behaviors (defined as reporting the occurrence of one or more extreme weight-control behaviors with concerning frequency).

## 3. Results

A total of 238 participants participated in the DEER study, out of which 217 participants (91.2%) completed all the required questionnaires and had no missing data for this study. We conducted a comparison between participants with complete data (*n* = 217) and those with missing data (*n* = 21) to assess differences in means and proportions of demographic characteristics. However, no significant differences were observed in any of the demographic characteristic variables. As a result, we employed a complete case analysis approach for the data analysis. 

When using a cutoff point of total EAT-26 score ≥ 12 points, 39% of participants (85 out of 217) were classified as being at a high risk of experiencing eating disorders. However, when using the traditional cutoff point of ≥20 points, 13% of participants (28 out of 217) were considered to be at high risk.

### 3.1. Descriptive Characteristics

Descriptive characteristics according to gender expression of participants are described in Table 1. Regarding differences related to gender expression, no statistically significant differences were observed between participants categorized as masculine or feminine/androgynous, except for relationship status. Among those self-described as feminine/androgynous, a higher proportion reported being single compared to those identified as masculine (70% vs. 54.7%, *p* = 0.03). 

### 3.2. Differences in Weight-Related Variables according to Gender Expression

There were no statistically significant differences in the means of current BMI, ideal BMI, and BMI discrepancy (the difference between current and ideal BMI). Similarly, there were no significant differences in the proportion of weight status categories (Table 2). It is worth noting that the majority of participants had an ideal BMI lower than their current BMI, resulting in a mean BMI discrepancy of 1.3 ± 3.2 kg/m^2^ in this population.

Interestingly, we observed a strong correlation between current BMI and BMI discrepancy (correlation coefficient = 0.90, *p* < 0.0001). This indicated that participants with higher BMIs tended to have a greater discrepancy between their current and ideal BMI compared to those with lower BMIs.

### 3.3. Association between Gender Expression, Weight Status, and Risk of Experiencing Eating Disorders

The associations between gender expression, BMI, and BMI discrepancy as predictor variables, and the total and subscale scores of EAT-26 as outcome variables are described in Table 3. Additionally, the associations between these predictor variables and the odds of experiencing a high risk for eating disorders and engaging in extreme weight-control behaviors are described in Figure 1. (Additional details can be found in Appendix A).

Regarding gender expression, after adjusting for other covariates, we found no significant association between gender expression and total EAT-26 scores. However, participants self-described as feminine/androgynous had 50% lower odds of experiencing a high risk for eating disorders compared to those self-described as masculine. Furthermore, participants self-described as feminine/androgynous had a 2.1-point lower score on the dieting subscale compared to participants self-described as masculine. No statistically significant associations were observed for the other subscale scores or the odds of engaging in extreme weight-control behaviors.

Regarding current BMI, when adjusting for other covariates, we found no significant association between BMI and total EAT-26 scores or the odds of experiencing a high risk for eating disorders. However, for the subscale scores, each 1-unit increase in BMI was associated with a 0.1-point increase in the bulimia and food preoccupation subscale score, but a 0.22-point decrease in the oral control subscale score. Additionally, each 1-unit increase in BMI was associated with 9% higher odds of engaging in extreme weight-control behaviors.

Similar to BMI, when adjusting for other covariates, BMI discrepancy showed no significant association with total EAT-26 scores or the odds of experiencing a high risk for eating disorders. However, for the subscale scores, each 1-unit increase in BMI discrepancy was associated with a 0.14-point increase in the bulimia and food preoccupation subscale score, and a 0.27-point decrease in the oral control subscale score. Additionally, each 1-unit increase in BMI discrepancy was associated with 16% higher odds of engaging in extreme weight-control behaviors.

## 4. Discussion

This study aimed to explore the relationship between gender expression, weight status, and risk of experiencing eating disorders among Thai gender-diverse adults assigned male at birth. The findings revealed that individuals who self-described as feminine or androgynous had 50% lower odds of experiencing a high risk for eating disorders. Additionally, higher BMI and BMI discrepancy were associated with higher odds of engaging in extreme weight-control behaviors.

Interestingly, the prevalence of experiencing a high risk for eating disorders in this small sample population of Thai gender-diverse adults assigned male at birth was 39% or 13%, depending on the cutoff point for the total EAT-26 used (12 points or more vs. 20 points or more, respectively). Notably, these prevalence rates aligned with other studies that utilized the EAT-26 in different populations. For instance, a study conducted on Hispanic gay men, defined as adults assigned male at birth who reported their gender identity as males and had sexual orientation towards males, reported a prevalence of 13% using the cutoff point at 20 points or more [21], while a study on college students in South India, 44% male between 18–21 years of age, reported a prevalence of 14% [26].

It is important to note that the generalizability of findings from this study could be limited as the majority of the participants had high educational attainment (bachelor’s degree or higher). Previous studies have shown that higher educational attainment is associated with an increased risk of experiencing eating disorders, particularly through higher traits of perfectionism that appeared to be increased with educational attainment [27,28]. Thus, the prevalence of eating disorder risk observed in this sample population could be substantially overestimated. However, the distribution of educational attainment status in gender-diverse adults assigned male at birth living in Thailand is not well-established as, to our knowledge, no formal survey has been conducted to obtain this specific data.

### 4.1. Association between Gender Expression and Weight Status

The findings from this study were consistent with previous research indicating that gender expression in individuals assigned male at birth, commonly reported as boys or men, is unlikely to be associated with weight status [29,30]. However, the majority of participants in this study expressed a desire for a lower weight, highlighting a potential discrepancy between individuals’ current and desired body image. Such discrepancy may contribute to body dissatisfaction and increased risk of experiencing eating disorders [31].

### 4.2. Association between Gender Expression and Risk of Experiencing Eating Disorders

Within the LGBTQ+ community, identifying the specific populations at higher risk has been challenging due to limited number of studies and the heterogeneity of the target population across studies [1]. In this study, participants who self-identified as feminine or androgynous demonstrated approximately a 50% lower likelihood of experiencing a high risk for eating disorders compared to those who identified as masculine. This association may be partly attributed to the lower scores observed in the dieting subscale. Interestingly, dieting behavior is typically perceived to be more prevalent among individuals identified as female, as they are more prone to experiencing body dissatisfaction related to their weight and shape compared to individuals identified as male [32]. However, consistent with the findings, previous studies showed that femininity serves as a protective factor for adults assigned male at birth in terms of disordered body image and eating behaviors [33].

### 4.3. Association between Weight Status and Risk of Experiencing Eating Disorders

This study showed that BMI was associated with specific subscale scores of the EAT-26, such as bulimia and food preoccupation subscale scores and oral control subscale scores. The lack of association between BMI and total EAT-26 score in this study may be due to the opposing effects of these two subscales, thereby canceling each other out. This highlights the heterogeneity of eating disorders, indicating that using total scores for EAT-26 may not effectively differentiate between different types of eating disorders.

However, we found that increased BMI is associated with higher odds of engaging in extreme weight-control behaviors, indicating a potential compensatory response to body weight dissatisfaction. This finding is supported by previous studies showing that higher body weights are associated with suboptimal eating behaviors and extreme weight-control behaviors [34,35,36], although the debate remains whether this is a result of personal responsibility or the perpetuation of weight stigmatization in society [37]. In fact, several longitudinal studies have demonstrated that adolescents and young adults, regardless of their assigned sex at birth, who engage in disordered eating and unhealthy weight-control behaviors, experience significant weight gain over time [38].

Similar to current BMI, BMI discrepancy was associated with a slight increase in the bulimia and food preoccupation subscale score and a decrease in the oral control subscale score, along with an increased likelihood of engaging in extreme weight-control behavior. The findings were consistent with previous studies suggesting a potential relationship between body dissatisfaction and extreme weight-control behaviors [39,40].

Interestingly, a strong correlation was observed between BMI and BMI discrepancy. While high BMI discrepancy can be observed across various weight levels (e.g., individuals with a BMI considered within the normal range may still exhibit a high BMI discrepancy if they desire a significantly lower weight), the findings suggest that individuals with higher body weight may experience more dissatisfaction with their body image and have a stronger desire to attain their ideal weight. These findings align with previous studies establishing a connection between body weight, body dissatisfaction, and the risk of experiencing eating disorders [41].

### 4.4. Strengths and Limitations

There were several notable strengths in this study. Firstly, it addresses a specific and underrepresented population in current research. To the best of our knowledge, no peer-reviewed study has been published that examined the risk of experiencing eating disorders and weight status among the gender-diverse population in Thailand, particularly focusing on individuals assigned male at birth. Thus, this study not only contributes significant findings but also highlights the need for further research, such as investigating the underlying mechanisms that drive the associations between weight status and the risk of experiencing eating disorders in Thai LGBTQ+ adults. Moreover, the study achieved a high completion rate, with 91.2% of participants completing all required questionnaires. This high rate enhances the reliability and validity of the findings. Additionally, the study employed multivariate and subscale analyses to provide a more nuanced understanding of the relationship between gender expression, weight status, and risk of experiencing eating disorders.

However, it is also important to consider the limitations of this study. Firstly, the small sample size and convenience-based sampling approach may limit the generalizability of the findings. The sample population consisted of young, educated individuals and may not fully represent the larger population of Thai gender-diverse adults assigned male at birth. Secondly, the cross-sectional design limits the ability to establish causal relationships between gender expression, weight status, and the risk of experiencing eating disorders. Thirdly, the use of self-reported measures introduces the possibility of recall bias. However, as we assume that both groups have the same degree of bias, this should not be a major concern when comparing within the sample population. Additionally, self-reported weight and height, while considered useful, are not considered gold-standard methods of anthropometric assessment. Fourthly, although the EAT-26 was validated and translated, it was not specifically validated in the LGBTQ+ population, as is the case with most instruments. Validation results were extrapolated from studies conducted with populations assigned female at birth. Moreover, while we utilized a validated tool to determine gender expression, we did not utilize specific tools to assess measures of drive for masculinity and femininity, which could provide further understanding to the observed findings. Lastly, there could be other risk factors for eating disorders, such as childhood trauma, a family history of eating disorders, or body image concerns, that were not included in this study.

## 5. Conclusions

This study provides preliminary insights into the relationship between gender expression, weight status, and risk of experiencing eating disorders among Thai gender-diverse adults assigned male at birth. The findings suggest that gender expression may influence the risk of eating disorders, with a more feminine/androgynous gender expression potentially being protective. Although no significant associations were found between weight status and overall risk of experiencing eating disorders, higher BMI and BMI discrepancy were associated with extreme weight-control behaviors. Further research, using longitudinal designs and more objective measures, is needed to expand the understanding of these complex relationships to effectively develop interventions to mitigate the risk of experiencing eating disorders in this population.

## Figures and Tables

**Figure 1 nutrients-15-03700-f001:**
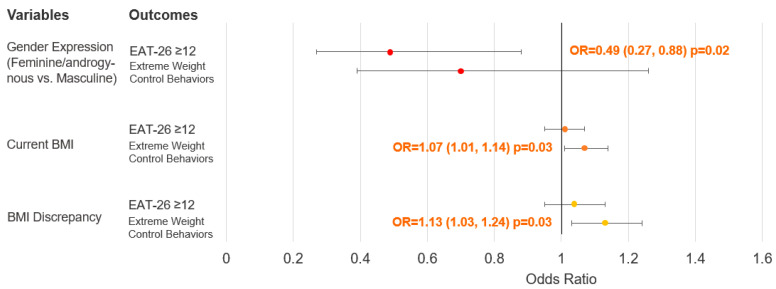
Multivariable logistics regression to explore gender expression and weight status in association with the risk of experiencing eating disorders and extreme weight-control behaviors in Thai gender-diverse adults assigned male at birth (*n* = 217). Abbreviation: BMI, body mass index; OR, odds ratio, EAT-26, Eating Attitude Test-26. Results are shown as odds ratio (95% confidence interval) and *p*-value.

**Table 1 nutrients-15-03700-t001:** Descriptive Characteristics of Thai Gender-diverse Adults Assigned Male at Birth who Participated in the DEER Study according to Gender Expression (*n* = 217).

Descriptive Characteristics	Total (*n* = 217)	t- or χ^2^ Value (*df*)	Gender Expression		*p*-Value
			Masculine (*n* = 137)	Feminine/Androgynous (*n* = 80)	
Age, years, mean (SD)	29.9 (6.19)	0.83 (215)	30.2 (6.02)	29.5 (6.49)	0.41 ^‡^
Birthplace, *n* (%)		0.05 (1)			0.82 ^§^
Bangkok	128 (59.0%)		80 (58.4%)	48 (60.0%)	
Other provinces	89 (41.0%)		57 (41.6%)	32 (40.0%)	
Income per month, *n* (%)		7.75 (3)			0.05 ^§^
Quartile 1 (THB 21,500 ^¶^ or less)	54 (24.9%)		27 (19.7%)	27 (33.8%)	
Quartile 2 (THB 21,501–33,000 ^¶^)	51 (23.5%)		30 (21.9%)	21 (26.3%)	
Quartile 3 (THB 33,001–60,000 ^¶^)	61 (28.1%)		44 (32.1%)	17 (21.3%)	
Quartile 4 (THB 60,001 ^¶^ or more)	51 (23.5%)		36 (26.3%)	15 (18.8%)	
Current Living status, *n* (%)		1.09 (2)			0.58 ^§^
Alone	92 (42.4%)		59 (43.1%)	33 (41.3%)	
With Friends/Family	92 (42.4%)		55 (40.1%)	37 (46.3%)	
With partner/couple	33 (15.2%)		23 (16.8%)	10 (12.5%)	
Religion, *n* (%)		1.08 (2)			0.58 ^§^
Atheist	69 (31.8%)		43 (31.4%)	26 (32.5%)	
Buddhist	142 (65.4%)		89 (65.0%)	53 (66.3%)	
Other religion	6 (2.8%)		5 (3.6%)	1 (1.3%)	
Educational attainment, *n* (%)		0.62 (1)			0.43 ^§^
Lower than bachelor’s degree	8 (3.7%)		4 (2.9%)	4 (5.0%)	
Bachelor’s degree or higher	209 (96.3%)		133 (97.1%)	76 (95.0%)	
Relationship Status, *n* (%)		4.91 (1)			0.03 ^§^
Single	131 (60.4%)		75 (54.7%)	56 (70.0%)	
Partnered/Married	86 (39.6%)		62 (45.3%)	24 (30.0%)	
Current Smoking Status, *n* (%)		0.94 (1)			0.33 ^§^
Yes	12 (5.5%)		6 (4.4%)	6 (7.5%)	
No	205 (94.5%)		131 (95.6%)	74 (92.5%)	
Current Alcohol Consumption, *n* (%)		2.88 (1)			0.09 ^§^
Yes	74 (34.1%)		41 (29.9%)	33 (41.3%)	
No	143 (65.9%)		96 (70.1%)	47 (58.8%)	
History of Mental Health Issues, *n* (%)		0.03 (1)			0.86 ^§^
Yes	26 (12.0%)		16 (11.7%)	10 (12.5%)	
No/unsure	191 (88.0%)		121 (88.3%)	70 (87.5%)	

All comparisons were made between masculine vs. feminine/androgynous group; ^‡^ Independent *t*-test *p*-value; ^§^ chi-square *p*-value; ^¶^ THB, Thai Baht; *df*, degree of freedom.

**Table 2 nutrients-15-03700-t002:** Differences in weight-related variables among Thai gender-diverse adults assigned male at birth according to gender expression (*n* = 217).

Weight-Related Variables	t- or χ^2^ Value (*df*)	Gender Expression		*p*-Value
		Masculine (*n* = 137)	Feminine/Androgynous (*n* = 80)	
Current BMI ^†^, kg/m^2^, mean (SD)	0.19 (122.7)	23.5 (4.01)	23.4 (5.86)	0.85 ^‡^
BMI ^†^ Category, *n* (%)	7.03 (4)			0.13 ^§^
Underweight		4 (2.9%)	7 (8.8%)	
Normal weight		64 (46.7%)	43 (53.8%)	
Overweight		34 (24.8%)	12 (15.0%)	
Obesity class I		27 (19.7%)	12 (15.0%)	
Obesity class II		8 (5.8%)	6 (7.5%)	
Ideal BMI ^†^, kg/m^2^, mean (SD)	1.22 (215)	22.3 (2.17)	21.9 (2.57)	0.22 ^§^
BMI ^†^ Discrepancy ^¶^, kg/m^2^, mean (SD)	−0.52 (119.5)	1.2 (2.65)	1.5 (4.04)	0.60 ^‡^

^†^ BMI, body mass index; ^‡^ independent *t*-test *p*-value; ^§^ chi-square *p*-value; ^¶^ BMI discrepancy is defined as the difference between current and ideal BMI (BMI_current_-BMI_ideal_); *df*, degree of freedom.

**Table 3 nutrients-15-03700-t003:** Multivariable linear regression to explore gender expression and weight status in association with total and subscale EAT-26 scores in Thai gender-diverse adults assigned male at birth (*n* = 217).

Variables	Total EAT-26 Score	Dieting Subscale	Bulimia and Food Preoccupation Subscale	Oral Control Subscale
Gender Expression				
Masculine	Ref.	Ref.	Ref.	Ref.
Feminine/Androgynous	−2.43 (1.08) *p* = 0.03	−2.43 (0.83) *p* = 0.004	0.16 (0.27) *p* = 0.54	−0.15 (0.40) *p* = 0.70
Current BMI ^†^	0.001 (0.11) *p* = 0.99	0.13 (0.09) *p* = 0.14	0.09 (0.03) *p* < 0.001	−0.22 (0.04) *p* < 0.001
BMI ^†^ discrepancy ^‡^	0.13 (0.16) *p* = 0.43	0.26 (0.13) *p* = 0.04	0.14 (0.04) *p* < 0.001	−0.27 (0.06) *p* < 0.001

^†^ BMI, body mass index; ^‡^ BMI discrepancy is defined as the difference between current and ideal BMI (BMI_current_-BMI_ideal_). Results are shown as beta coefficient (standard error) and *p*-value.

## Data Availability

Data associated with the paper is available upon request.

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
