# Peer review of "Gender Expression, Weight Status, and Risk of Experiencing Eating Disorders among Gender-Diverse Adults Assigned Male at Birth in Bangkok, Thailand"

_nutrients, 2023, doi:10.3390/nu15173700_

Round 1

Reviewer 1 Report

The paper reports data from an online survey of people from the LGBTQ community assigned male sex at birth. Participants  who describe themselves as “feminine” or “androgynous” are compared with participants who describe themselves as “masculine”. In this cross-sectional study, regression analyses identified a lower risk for eating disordered behavior in participants who describe themselves as “feminine” or “androgynous”.

The paper addresses a relevant research question and the data are interesting. However, there are concerns regarding the language, the rationale and the number of analyses of the study.

Major points needing the attention of the authors are:

1) There is an annoying confusion regarding the use of present tense and past tense in the paper. As a general rule, all that was done in the past should be reported in the past tense, i. e. the general description of the study and the report of the results. The present tense should be used when referring to elements of the paper, for example in the description of the statistical methods and the reference to tables or figures.

2) The participants of this study are people from the LGBTQ community who describe their gender identity as different from the sex assigned at birth. The authors claim these participants as “marginalized individuals”. This construct needs clarification, describing who puts the participants on the margin and which disadvantages for the participants emerge from this marginalization. The authors may consider to avoid the terms “marginalization” or “marginalized individuals”, as this construct does not contribute to the research question.

The sentences in lines 60-63 on page 2 seem to include contradictory statements. Please clarify.

3) The design of the study compares two subgroups of the sample. This should be more clearly described in the methods.

4) A major weakness of the study is the very substantial bias regarding the education of the participants. The large majority of the participants had a bachelor degree. One may hypothsize that this is a consequence of the self-selection procedure of the study and the readiness of the participants to reveal the gender orientation. How large is this percentage in the overall LGBTQ community and in the Thai population? Please discuss this limitation more elaborately and in detail in a separate paragraph.

5) No rationale is given for the inclusion of covariates in part of the regression analyses. Considering the rather skewed distribution of some covariates and the large number of analyses in the small self-selected sample size (238 of 8 millions?), I would recommend to remove the regression analyses with adjusted covariates, as well as the regression analyses on the subscales of the EAT-26. Report only unadjusted results for the EAT-26 total score and include the explained variance of the model.

6) For all statistical tests please report all statistics, not just p-values. The sentence on page 4, line 195 is unprecise. Please report the average age of the total sample in the same format as other means and standard deviations. Remove the word approximately as you can report exact values.

7) Results of the regression analyses are reported in a table (linear regression) and in a figure (logistic regression). While the figure is fine, I would recommend to add a table with the results of the logistic regression.

8) The discussion is too long and needs more more focus. Do not repeat results in the discussion. In the paragraph on page 8, lines 268 – 274, please add a short description of the definition of gay men and the characteristics of college students. The paragraph on page 8, lines 289 – 294 includes several speculative statements, and I would recommend to remove this paragraph from the discussion. The next paragraph (page 8, lines 295 – 311 lacks references and includes much repetition of results. I would  also recommend to remove the paragraphs on page 9, lines 312 – 322, they do not contribute new aspects to the discussion.

9) The limitations of an online survey (e. g. self-selection, self-report) are mentioned in the limitations. The authors evidently did some verification of the participants, but it is unclear if this was done for all participants in a systematic way (desirable) or only for participants with unclear of missing data. As mentioned above, the severe educational bias needs a more elaborate discussion. The authors claim a high completion rate of more than 90%, but this refers only to participants of the survey. From what is reported in the paper, only a very small proportion of the LGBTQ community participated in the survey, limiting the power of the study.

Measures of drive for masculinity were not included in the study and this should be added as a limitation.

Please see my comments on the paper.

Author Response

I sincerely appreciate the chance to revise and resubmit the manuscript titled "nutrients-2535485". The provided comments proved to be both enlightening and invaluable as we endeavored to elevate the manuscript's overall quality. We have diligently integrated the majority of the suggestions provided by the reviewers. Detailed responses to each comment are presented in the subsequent sections. Notably, all references to line numbers in the forthcoming sections are aligned with the revised version of the manuscript. Kindly inform me if any additional questions or concerns arise. We remain hopeful that this revised rendition of the manuscript will duly fulfill the publication criteria.

  1. There is an annoying confusion regarding the use of present tense and past tense in the paper. As a general rule, all that was done in the past should be reported in the past tense, i. e. the general description of the study and the report of the results. The present tense should be used when referring to elements of the paper, for example in the description of the statistical methods and the reference to tables or figures.

Response: We have thoroughly reviewed the entire manuscript and rectified grammatical inaccuracies and discrepancies in tense usage as appropriate.

  1. The participants of this study are people from the LGBTQ community who describe their gender identity as different from the sex assigned at birth. The authors claim these participants as “marginalized individuals”. This construct needs clarification, describing who puts the participants on the margin and which disadvantages for the participants emerge from this marginalization. The authors may consider to avoid the terms “marginalization” or “marginalized individuals”, as this construct does not contribute to the research question. The sentences in lines 60-63 on page 2 seem to include contradictory statements. Please clarify.

Response: We removed the contradictory statement and replaced it with a new statement that should improve clarity (Line 62 – 63). In adopting the umbrella term "gender marginalized," our intention was to cover the entirety of gender identities beyond cisgender heterosexuality. However, as we recognize the limitations of this term, we replaced the term with “gender diverse” to preserves the essence of describing gender identities beyond cisgender heterosexuality, including non-binary terms, as participants in this study self-identified as gays, bisexuals, transgenders, queers, and other non-binary genders (e.g., pansexual, asexual, etc.)

  1. The design of the study compares two subgroups of the sample. This should be more clearly described in the methods.

Response: We added a sentence describing the study design in details. (Line 87 – 90).

  1. A major weakness of the study is the very substantial bias regarding the education of the participants. The large majority of the participants had a bachelor degree. One may hypothesize that this is a consequence of the self-selection procedure of the study and the readiness of the participants to reveal the gender orientation. How large is this percentage in the overall LGBTQ community and in the Thai population? Please discuss this limitation more elaborately and in detail in a separate paragraph.

Response: We acknowledge this substantial bias and added a discussion regarding educational attainment and risk of experiencing eating disorders as a separate paragraph (Line 274 – 282).

  1. No rationale is given for the inclusion of covariates in part of the regression analyses. Considering the rather skewed distribution of some covariates and the large number of analyses in the small self-selected sample size (238 of 8 million?), I would recommend to remove the regression analyses with adjusted covariates, as well as the regression analyses on the subscales of the EAT-26. Report only unadjusted results for the EAT-26 total score and include the explained variance of the model.

Response: We removed adjusted regression analyses and only report the unadjusted model as suggested. We also revised the abstract and replaced the adjusted with unadjusted odds ratio. Regarding subscale analyses, we believe that the subscale scores provided meaningful results as they allowed us to differentiate sub-characteristics of eating disorder risks (e.g., bulimia and food preoccupation vs. food restriction) that added to the understanding of the results when compared to reporting only the total EAT-26 scores. Thus, we would like to retain the subscale analyses in this study.

  1. For all statistical tests please report all statistics, not just p-values. The sentence on page 4, line 195 is unprecise. Please report the average age of the total sample in the same format as other means and standard deviations. Remove the word approximately as you can report exact values.

Response: We reported beta-coefficient and standard error for linear regression analyses and reported odds ratio with 95% confidence intervals for logistic regression analyses. We removed the word approximately in the sentences from Line 195 – 198. We removed the entire description of demographic characteristics in the text as it repeated the results shown in Table 1.

  1. Results of the regression analyses are reported in a table (linear regression) and in a figure (logistic regression). While the figure is fine, I would recommend to add a table with the results of the logistic regression.

Response: The table for logistic regression was originally submitted as Supplementary Table S2. We revised Figure 1. To comply with changes in Response 5 and added a sentence to mention about the Supplementary Table S2. (Line 225 – 226).

  1. The discussion is too long and needs more focus. Do not repeat results in the discussion. In the paragraph on page 8, lines 268 – 274, please add a short description of the definition of gay men and the characteristics of college students. The paragraph on page 8, lines 289 – 294 includes several speculative statements, and I would recommend to remove this paragraph from the discussion. The next paragraph (page 8, lines 295 – 311 lacks references and includes much repetition of results. I would also recommend to remove the paragraphs on page 9, lines 312 – 322, they do not contribute new aspects to the discussion.

Response: We added a description of the gay men and characteristics of the college student in cited studies (Line 270 – 282). We revised the whole discussion section to shorten the length and make the discussion more focused.

  1. The limitations of an online survey (e. g. self-selection, self-report) are mentioned in the limitations. The authors evidently did some verification of the participants, but it is unclear if this was done for all participants in a systematic way (desirable) or only for participants with unclear of missing data. As mentioned above, the severe educational bias needs a more elaborate discussion. The authors claim a high completion rate of more than 90%, but this refers only to participants of the survey. From what is reported in the paper, only a very small proportion of the LGBTQ community participated in the survey, limiting the power of the study. Measures of drive for masculinity were not included in the study and this should be added as a limitation.

Response: We added a paragraph discussing the limited generalizability of the findings due to educational bias (line 274 – 278) and a sentence describing the lack of measures of drive for masculinity (and femininity) (Line 358 – 361).

Reviewer 2 Report

Dear Authors this a very interesting study on a subject that has not been studied previously.

The Introduction provides the relevant scientific knowledge and the literature gaps, the Methodology describes the sampling procedure and the methods used, while the Results are clearly presented, the Discussion comments on the findings and the strengths and limitations of your study and finally the Conclusions are in line with the Results.

There are some minor problems in the text: some references are not within brackets, while at lines 225 and 227-228 an error message appears.

Author Response

We sincerely appreciate the chance to revise and resubmit the manuscript titled "nutrients-2535485". The provided comments proved to be both enlightening and invaluable as we endeavored to elevate the manuscript's overall quality. We have diligently integrated the majority of the suggestions provided by the reviewers. Detailed responses to each comment are presented in the subsequent sections. Notably, all references to line numbers in the forthcoming sections are aligned with the revised version of the manuscript. Kindly inform me if any additional questions or concerns arise. We remain hopeful that this revised rendition of the manuscript will duly fulfill the publication criteria.

  1. There are some minor problems in the text: some references are not within brackets, while at lines 225 and 227-228 an error message appears.

Response: We revised the entire manuscript to make sure that all references are correctly cited and are in the correct format.

Reviewer 3 Report

The authors present a study on the risk of eating disorders in a population of gender marginalized adults assigned male at birth living in Bangkok, Thailand. This subject is little developed in the literature, which makes it an interesting topic to explore. The method is well detailed and clear, and corresponds to the objectives of the article.

Page 3 line 136, the validation of EAT26 should be less detailed, as it doesn't add anything important.

On page 5, line 198, income should be compared with the average wage in Thailand and not expressed in USD.

Results :

Table 1 is very detailed, but the text needs to be simplified, as it repeats many of the elements in the table.

What's the point of detailing religion if there's no analysis or commentary? This part should be removed.

On page 6, lines 225 and 227, there is an editing problem.

The basis of the association of feminine and androgynous gender expression should be explained in analyses. How do the authors think this group is homogeneous compared to male gender expression? Were any differences sought?

References :

Check reference 17

Author Response

We sincerely appreciate the chance to revise and resubmit the manuscript titled "nutrients-2535485". The provided comments proved to be both enlightening and invaluable as we endeavored to elevate the manuscript's overall quality. We have diligently integrated the majority of the suggestions provided by the reviewers. Detailed responses to each comment are presented in the subsequent sections. Notably, all references to line numbers in the forthcoming sections are aligned with the revised version of the manuscript. Kindly inform me if any additional questions or concerns arise. We remain hopeful that this revised rendition of the manuscript will duly fulfill the publication criteria.

  1. Page 3 line 136, the validation of EAT26 should be less detailed, as it doesn't add anything important.

Response: We acknowledged that the description of EAT-26 was very detailed. While we believe that the tool should still be valuable for use in this study, we would like to declare upfront for the readers that the Thai version of the tool was validated in Thai adults assigned female at birth, not the gender diverse population assigned male at birth. Thus, we slightly revised the paragraph for clarity, but would still like to retain the majority of the information. (Line 143 – 151)

  1. On page 5, line 198, income should be compared with the average wage in Thailand and not expressed in USD.

Response: Please see Response 3 below.

  1. Results: Table 1 is very detailed, but the text needs to be simplified, as it repeats many of the elements in the table.

Response: We removed detailed information that was repetitive with the information in Table 1. Thus, we also removed the description of the average wage (Response 2) as we did not discuss this in detail in the discussion section (as it was less relevant to the research question).

  1. Results: What's the point of detailing religion if there's no analysis or commentary? This part should be removed.

Response: We removed the sentence describing participants’ religious status.          

  1. Results: On page 6, lines 225 and 227, there is an editing problem.

Response: We corrected the citation. The text should refer to Table 3 and Figure 1.

  1. Results: The basis of the association of feminine and androgynous gender expression should be explained in analyses. How do the authors think this group is homogeneous compared to male gender expression? Were any differences sought?

Response: We added a paragraph to provide a rationale for combining the feminine with the androgynous group (Line 157 – 161).

  1. References: Check reference 17

Response: We added the author’s name for reference 17. The author’s name was labeled as “WHO Expert Consultation” for this article (https://pubmed.ncbi.nlm.nih.gov/14726171/). We also review other references to make all citations were correct.

Round 2

Reviewer 1 Report

The authors addressed most of my concerns and improved the paper considerably. I thank them for their close attention to my concerns. Supplementary Table S2 was not accessible to me and I cannot comment on this table. One point not addressed adequately by the authors is the reporting of statistical tests. I commented that for all statistical tests all statistics, not just p-values, should be reported. This is a generally accepted standard of scientific reporting. In their answers, the authors only referred to the reporting of the results of the regression analyses, and I wonder if there was a misunderstanding. The requirement of complete statistical reporting refers also to t- and chi-square tests. Please add information on t- or chi-square values, as well as degrees of freedom in Tables 1 and 2 as well as at some places in the text.

Author Response

Thank you sincerely for affording us the opportunity to revise and resubmit the manuscript titled "nutrients-2535485." In response to the feedback provided by Reviewer 1, specifically addressing the need to incorporate information pertaining to t- or chi-square values, coupled with degrees of freedom, within both Tables 1 and 2, as well as intermittently within the narrative, we have diligently attended to this observation. We have duly accommodated this requirement by introducing an additional column within both Table 1 and Table 2, effectively presenting the t- and chi-square values alongside their corresponding degrees of freedom (as exemplified in Line 206 and Line 219).

Upon comprehensive reassessment of the manuscript, we are inclined to posit that the elucidation of t- and chi-square values as presented within the tables themselves serves as an adequate explication. Hence, we refrained from incorporating these elucidations within the textual content. Furthermore, for enhanced clarity and reference, we kindly draw your attention to Supplementary Table S2 provided below.

Supplementary Table S2. Multivariable logistics regression to explore gender expression and weight status in association with the risk of experiencing eating disorders and extreme weight control behaviors in Thai gender diverse adults assigned male at birth (n=217)

Variables

EAT-26 ≥ 12

EAT-26 ≥ 20

Extreme Weight Control Behaviors

Gender Expression

     Masculine

Ref.

Ref.

Ref.

     Feminine/androgynous

0.49 (0.27, 0.88)

0.33 (0.12, 0.91)

0.70 (0.39, 1.26)

Current BMI

1.01 (0.95, 1.07)

1.00 (0.92, 1.09)

1.07 (1.01, 1.14)

BMI discrepancy§

1.04 (0.95, 1.13)

1.03 (0.92, 1.15)

1.13 (1.03, 1.24)

BMI, body mass index; EAT-26, Eating Attitute Test-26; §BMI discrepancy is defined as the difference between current and ideal BMI (BMIcurrent – BMIideal); Results are shown as odds ratio (95% confidence interval).
